# The hallmarks of childhood abuse and neglect: A systematic review

Jason Lang[1,2], Daniel M. Kerr[1]*, Papoula Petri-Romão[1], Tracey McKee[3], Helen Smith[3], Naomi Wilson[3], Marianna Zavrou[3], Paul Shiels[1]☯, Helen Minnis[1]☯

1 University of Glasgow, Glasgow, United Kingdom, 2 NHS Lanarkshire, Lanarkshire, United Kingdom, 3 NHS Greater Glasgow and Clyde, Glasgow, United Kingdom

☯ These authors contributed equally to this work.
* Daniel.Kerr@glasgow.ac.uk

**Data Availability Statement:** All relevant data are within the manuscript and its Supporting Information files.

**Funding:** The authors received no specific funding for this work.

## Abstract

### Background

Studies on the impacts of child maltreatment (CM) have been conducted in diverse areas. Mechanistic understanding of the complex interplay between factors is lacking. Hallmarking is an approach which identifies common factors across studies and highlights the most robust findings.

### Objectives

In a review of systematic reviews and meta-analyses, we addressed the following questions: 1) What are the hallmarks associated with exposure to CM across the bio-ecological spectrum? 2) What is the strength of evidence to support each hallmark? 3) What are the gaps that future research should address?

### Methods

A comprehensive literature search was carried out to find relevant systematic reviews or meta-analyses. 269 articles were read in full and 178 articles, encompassing more than 6000 original papers, were included in the final synthesis. All reviews were independently rated for quality by at least 2 reviewers using AMSTAR-2.

### Results

Of 178 review articles, 6 were rated as high quality (all meta-analyses) and 46 were rated as medium quality. Most were from high income countries.

### Conclusions

Based on the most commonly reported high-quality research findings we propose that the hallmarks of exposure to child maltreatment are: *Increased risk of psychopathology; Increased risk of obesity; Increased risk of high- risk sexual behaviours, Increased risk of smoking;* and *Increased risk of child maltreatment in children with disabilities.* Research gaps include a lack of focus on complexity and resilience. Little can be concluded about

**Competing interests:** The authors have declared no competing interests exist.

directions of causality or mechanisms. Adequately powered prospective studies are required to move the field forward.

## Introduction

Child maltreatment (CM) is common worldwide [1,2]. A large volume of research has explored the correlates of CM across the bio-ecological spectrum [3], from epigenetic changes [4], dysregulation of the immune system [5] and the hypothalamo-pituitary-adrenal (HPA) axis [6], through to the social factors conferring resilience against the impacts of CM [7]. Studies have consistently shown CM to be associated with a range of adverse physical, psychological and social outcomes [2], yet virtually nothing is known about the interplay between these disparate factors across the life-course, nor about how they interact with CM to result in these negative outcomes. This vast research base has therefore had limited impact on the ability of society to prevent exposure to CM or, when exposure has already occurred, to break the chain linking CM exposure to adverse outcomes [8].

A fuller understanding of the interplay between these wide-ranging factors will be necessary if we are to find ways to prevent CM and enhance resilience in those already exposed. Resilience is thought to result from complex and dynamic processes of adaptation to stressors that involve the activation of a variety of protective factors [7] operating at every level of the bio-ecological spectrum [4,9–11]: from genetics [4], through the HPA axis [9] and immune system [10], the brain [12], and into the family [13] and wider community [14], with the potential for reciprocal influences at all levels [7,15].

Much extant work in this field is based on retrospective reports of CM exposure from adults exposed in childhood. There is a relative paucity of high-quality longitudinal prospective studies beginning in childhood [5]. This is particularly concerning since there is poor agreement between retrospective and prospective reports of CM [16]. By their very nature studies of the effects of CM cannot be randomised, so are vulnerable to bias. Variations in the classification of both CM and outcomes across studies, differences in reporting, and different approaches to adjustment for confounding variables, all contribute to conflicting and sometimes confusing conclusions in the field. Furthermore, in interpreting research in this area it is important to distinguish the impacts of CM from the related but much wider concept of Adverse Childhood Experiences (ACEs), which, whilst including CM, also includes distinct exposures such as household dysfunction [8].

Due to the increasing diversity of outcomes under examination, and the wide range of quality in the published literature, it has become important to find a better way to conceptualise and integrate this broad evidence base. Doing so would enable researchers to better understand what evidence can be relied upon, what is known about the likely causes and outcomes of CM, how these might interact, and what this can tell us about likely mechanisms. This is where the concept of "hallmarking" might be useful. The hallmarking technique was first applied to cancer studies at a time when this literature was also experiencing a significant growth in volume and complexity [17]. The purpose was to find common factors by seeking commonalities across different studies and across most (if not all) types of cancer.

Applying this concept to the study of CM (which we define as child abuse—physical, emotional, and sexual; and neglect), we want to identify hallmarks across the entire biopsychosocial environment of the child and to consider the volume and quality of evidence for each of these hallmarks. Recent theoretical models have focussed on the human stress response system as the "control centre" for human adaptation to severe stresses such as abuse and neglect and

suggest that only a truly integrated approach involving all bio-ecological levels has the potential to identify mechanisms [15]. Some previous hallmarking processes have examined commonalities across both humans and other species [17] but we did not think that was appropriate here: whilst there are animal models of early life stress, we chose to look more specifically at CM as opposed to early life stress more broadly. Animal models cannot distinguish these.

Many thousands of papers have been written about factors associated with CM and many literature reviews have been conducted exploring these. In order to bring together such a large body of literature, we have conducted a 'review of reviews' [18] as the first stage of our hall-marking process, followed by a synthesis of the findings of these with reference to the bio-ecological model. We aimed to answer the following questions:

1. What does the literature identify as hallmarks of exposure to CM across the bio-ecological spectrum?

2. What is the strength of evidence to support each hallmark so identified?

3. What are the research gaps in this field, in terms of areas where further research, or better-quality research, is needed?

## Methods

The systematic review was performed in accordance with PRISMA guidelines [19]. Our PRIMSA checklist is available in S1 File. Studies were identified by searching the following electronic databases from 2009 to present: Ovid Medline ALL (R) (1946 to Present), OVID Embase Classic & Embase (1947 to Present), OVID PsycInfo (1806 to Present) and the Cochrane Database of Systematic Reviews. All searches were run on 29th May 2019.

The search strategy was developed by a Subject Specialist Librarian in consultation with the review group. The final draft Medline search strategy was peer reviewed by another librarian not involved in the review. The search strategy utilised a combination of subject headings and keywords; the strategy was adapted to each database as required to take account of differences in subject headings and search tools. Due to time constraints a systematic review search filter was applied to the search strategy to maximise specificity. The search filters were developed by the Health Information Research Unit at McMaster University, Canada [20–22]. In addition, the results were limited to English Language and, because we wanted to focus on the recent literature, more likely to evidence current theoretical models, a publication date limit was set of within the last ten years (2009 to May 2019). The master search strategy for OVID Medline ALL (R) can be found in S2 File.

The search strategy consisted of eight individual concepts drawn from the review question; these were searched individually and then combined to find relevant studies. The first search concept was 'child abuse & neglect' and the search terms included child abuse, childhood sexual abuse, child neglect and adverse childhood experiences. The second search concept was 'social factors', the search terms included socioeconomic factors, poverty, gender, sexuality, educational status and social support. The third search concept was 'genetic phenomena' and the search terms included genetics, epigenetics and biomarkers. The fourth concept was 'mental health', search terms included mental disorders, suicide, depression and PTSD. The fifth concept was 'physical health' and the search terms included obesity, smoking, heart disease and diabetes. The sixth search concept was 'stress responsivity'; the search terms included autonomic nervous system, stress response and heart rate. The seventh search concept was 'neuro-anatomical factors', the search terms include neuroimaging and MRI. The final search

concept was 'inflammatory/endocrine markers', the search terms included endocrine and immune biomarkers.

The PRISMA flow diagram [19] is shown in Fig 1. The total number of articles returned from the original search was 2255 and following removal of duplicates 1433 articles remained. 1433 records underwent title and abstract review for inclusion using inclusion/exclusion

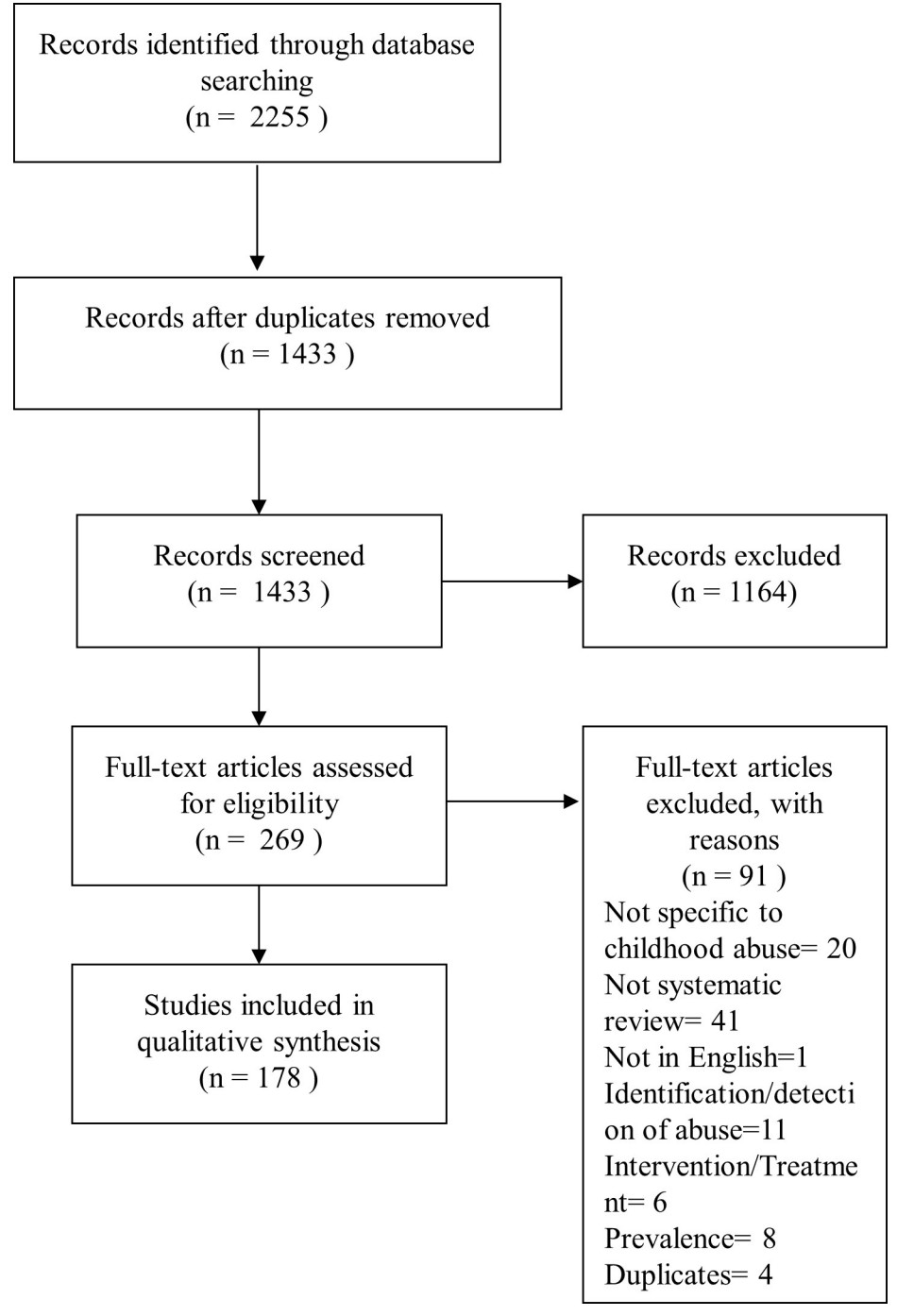

**Fig 1. PRISMA flow diagram.**

| Inclusion Criteria | Exclusion Criteria |
| --- | --- |
| Systematic Review or Meta-analysis | Original Articles, Conference Articles, Abstracts, Posters |
| Documented abuse or neglect (Physical abuse, emotional abuse, sexual abuse, neglect, or a combination) | No documented abuse. |
| Measures a potential cause or consequence of abuse or neglect | Studies purely concerned with the prevalence, detection, prevention or treatment of abuse which do not consider potential causes or consequences. |
| Human Participants | Animal Studies |
| Quantitative Studies (can be mixed qualitative / qualitative) | Qualitative Studies |
| Published since 2009 | Published before 2009 |
| English Language | Not English Language |

**Fig 2. Inclusion and exclusion criteria.**

criteria agreed prior to the search process, by at least 2 raters (see Fig 2 for inclusion/exclusion criteria). Where conflict existed, this was resolved in a conference of the authors. Following this process, 269 articles were read in full by at least two reviewers. A further 91 records did not meet inclusion criteria when read in full and were excluded (reasons in Fig 1). All exclusions where checked and agreed by at least two authors. This left 178 articles which were data extracted and rated for quality using the AMSTAR-2 checklist [23]. AMSTAR-2 is a structured tool for critical appraisal of systematic reviews and meta-analyses of randomised or non-randomised studies. It consists of 16 items including design of the review, search strategy, study selection, risk of bias assessment and synthesis of findings (see S3 File for all items). For quality scoring purposes AMSTAR-2 deems 7 items to be critical: protocol registration before commencement of the review; adequacy of the literature search; justification for excluding individual studies; risk of bias from individual studies being included in the review; appropriateness of meta-analytic methods; consideration of risk of bias when interpreting the results of the review; and assessment of presence and likely impact of publication bias. AMSTAR-2 assigns quality scores to studies ranging from high to critically low. High-quality studies require no or one non-critical weakness. All articles were independently rated by at least two authors and discrepancies resolved at conference. Data on study setting, type of abuse, number of studies, and results were extracted.

We considered quantitative analysis using a network approach, but this was not possible due to the wide range of variables examined in the reviews. Instead results are presented in narrative format.

## Results

### Characteristics of studies

178 studies were included. 43% of these studies were meta-analyses (n = 77) and 57% were systematic reviews (n = 102).

The review included studies from North America, Europe, South America, Asia, Africa and Australasia. Not all studies identified the sources of the studies (n = 14), and many reviews and

meta-analyses included studies from more than one country. Some studies used phases such as 'non-US high income countries' or 'industrialised countries', others grouped countries by continent, albeit not consistently. However more than 80% were from WEIRD (Western, educated, industrialised, rich and democratic) countries.

We conceptualised and organised our findings using an adapted version of the ecological model developed by Bronfenbrenner [3]. This conceptualises developmental processes as an interaction between the child and its environment on several levels (Fig 3). These are the micro-, meso-, exo-, and macrosystem. The microsystem encompasses the child, including their biology and the family relationships. The mesosystem describes the environment, relationships just out of the nuclear home, i.e. friendships, school, and extended family. Following that, the exosystem includes the interaction between the child and the wider neighbourhood and community. Lastly the Macrosystem, embeds the child society, recognising policy, religion, and wider structures. In this study, the search terms have been mapped to Bronfenbrenner's ecological model. However, more levels were added within the microsystem to reflect the different areas of research. Therefore our results have been grouped as: 'Biochemical Factors', 'Genes and Epigenetic Factors', 'Mind and Body', and 'Social factors'. *'Biochemical factors'* included studies of markers of inflammation, the immune system, cortisol and other biomarkers. *'Genes and Epigenetic factors'* looked at genetic and epigenetic markers. Due the number of papers in the categories *'Body and Mind'* and *'Social Factors'*, these were further divided into subthemes. *Body and Mind*: Mental Health and Substance Use/Misuse; Physical Health; Brain structure, neurodevelopment, cognition and personality; *Social Factors*: Relationships, parenting, sexual behaviour and Offending and antisocial behaviour. The distribution of papers by category is shown in Table 1.

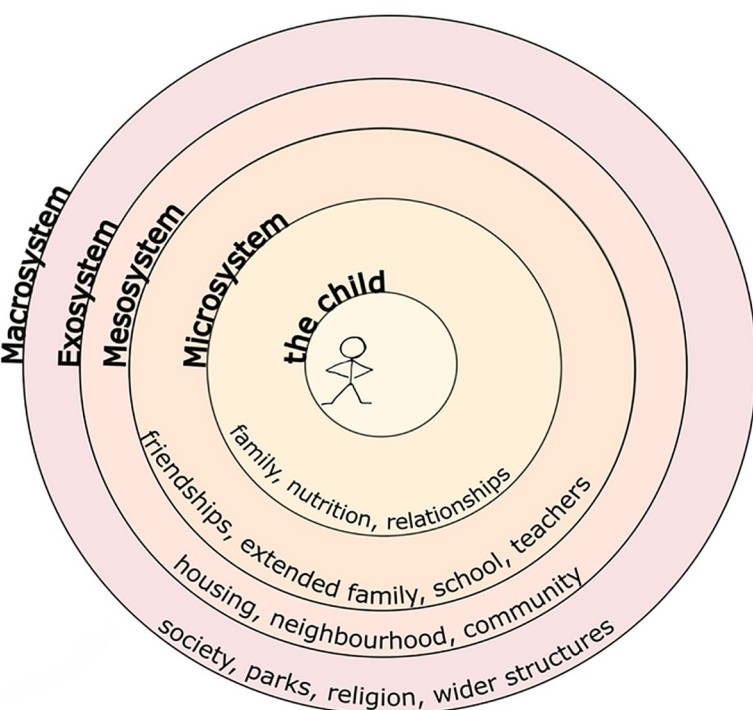

**Fig 3. Bio-ecological model (Adapted from Bronfenbrenner 2005 [3]).**

**Table 1. Types and numbers of papers sorted by thematic category and subthemes.**

| Category | Subtheme | Bio-Ecological Levels | Number Of Papers |
|---|---|---|---|
| • Biochemical Factors | | Microsystem | 7 |
| • Genes and Epigenetic Factors | | Microsystem | 11 |
| • Mind and Body | | | |
| | Individual Mental Health, Substance Use and Misuse | Microsystem | 74 |
| | Brain Structure, Neurodevelopment, Cognition and Personality | Microsystem | 18 |
| | Physical Health | Microsystem | 17 |
| • Social Factors | | | |
| | Environmental Risk Factors | Exo- & Macrosystem | 18 |
| | Offending and Antisocial Behaviour | Meso-, Exo- & Macrosystem | 10 |
| | Relationships, Parenting, Sexual Behaviour | Mesosystem | 16 |
| | Education/Adults Economic Status | Meso-, Exo- & Macrosystem | 2 |
| | Resilience Factors | All Systems | 6 |

142 studies (79%) investigated a combination of types of abuse and neglect, whereas 33 (18%) concentrated on sexual abuse, 2 (1%) on physical abuse and 3 (2%) on emotional abuse. The modal number of included studies per review was 12 (range 2–393).

## Quality of the studies

Overall, over two thirds (70%, n = 126) of papers were rated as low or critically low quality. Just over a quarter (27%, n = 46) were rated as moderate quality and only 3% (n = 6) were rated as high-quality. Most lower ratings could be explained by the lack of a risk of bias assessment or a failure to incorporate such an assessment into the synthesis. Further, many papers lacked a rigorous search strategy and data extraction procedure.

No systematic reviews achieved high-quality rating; however, 6 meta-analyses did. Fig 4 shows the quality rating of articles by review type and thematic category.

## Findings of high-quality studies

Six papers received a high-quality AMSTAR-2 rating [24–29]. All were meta-analyses. Five of these papers fell into the thematic category of Mind and Body and one into the category of Social Factors. Of the five papers in the Mind and Body category, four had the subtheme of 'mental health and substance use'. All these papers investigated more than one type of abuse. Details of these papers are summarised in Table 2.

Bailey et al. [24] studied the association between childhood trauma and severity of hallucinations and delusions in psychotic disorders. Their review included 41 studies, of which 29 were included in the meta-analysis with 4680 participants in total. This review defined childhood trauma to include sexual abuse, physical abuse, emotional abuse, physical neglect, emotional neglect, and bullying. The countries of origin of the included studies were not stated, however only studies published in English were included. They found that childhood sexual abuse and neglect was significantly correlated with severity of hallucinations (r = .172, p<0.001). Sexual abuse and physical or emotional neglect was also associated with delusion severity (r = .199, p<0.001). Further, sexual abuse increased severity of positive symptoms, and negative symptoms of schizophrenia were associated with childhood neglect.

Castellvi et al. [25] investigated the association between exposure to violence and risk for suicide. The meta-analysis included 26 papers with a total sample of 143,730. Violence

| Meta-analysis total | N=76 | Systematic Review | N=102 |
|---|---|---|---|
| High Quality | n=6 (7.9%) | High Quality | n=0 (0%) |
| Medium Quality | n=22 (28.9%) | Medium Quality | n=24 (23.5%) |
| Low Quality | n=48 (63.1%) | Low Quality | n=78 (76.5%) |

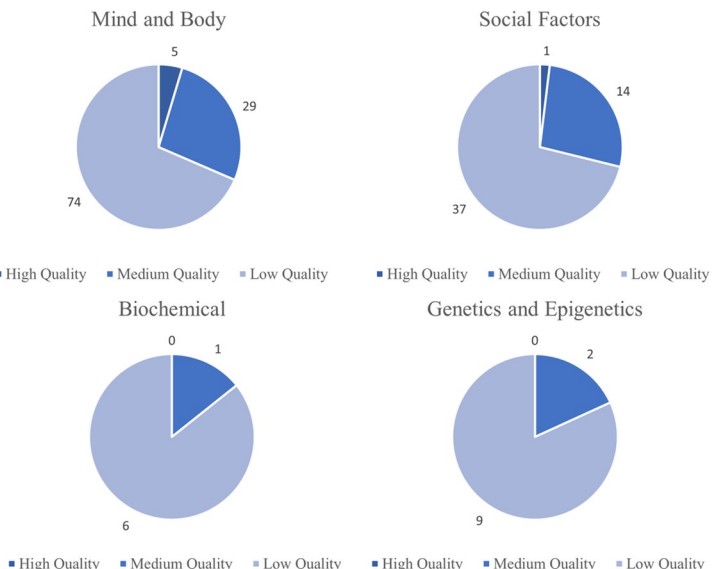

**Fig 4. Overview of AMSTAR-2 ratings by type of review and theme.**

was defined as child maltreatment, bullying, dating violence and community violence. The included studies originated in the Netherland, New Zealand, United States, Norway, Canada, United Kingdom, Denmark, Finland. They found participants with experience of physical abuse to have an increased risk of suicidal behaviour (OR = 2.25; 95% CI: 1.85–2.73). The evidence was weaker for the association between sexual abuse and suicide behaviour. There were not enough studies that investigated the link between emotional abuse and suicide behaviour. The association between neglect and suicide behaviour was not significant.

Fusar-Poli et al [26] performed a systematic review and meta-analysis of environmental factors associated with ultra-high risk for psychosis, including childhood abuse and neglect. Forty-four studies were included in their review. This review only included papers written in English. They found strong evidence that emotional abuse (OR = 5.843, 95% CI 1.794–19.027) and physical neglect (OR = 3.066, 95% CI 1.043–9.013) experienced during childhood are associated with ultra-high-risk state for psychosis.

Jones et al. [27] reviewed the risk of violence against children living with disabilities. Of the 17 papers that are included in the meta-analysis, 11 included risk estimates and 16 included prevalence rates of violence exposure. The sample sizes were 13,505 children and 14,721 children, respectively. Violence was defined as physical violence, sexual violence, emotional abuse, neglect and any combination of those. They found that children with disabilities were at increased risk of abuse and neglect in comparison to non-disabled children (OR = 3·68, 95% CI 2·56–5·29). The pooled prevalence of violence against children with disabilities was 26.7% (95% CI 13.8–42.1); this analysis did not include a control group to allow comparison of the

**Table 2.**

| Author | Year | Category | AMSTAR score | Type of abuse investigated | Number of included studies | Number of participants in meta-analysis | Countries of studies | Summary of result |
|---|---|---|---|---|---|---|---|---|
| Bailey T | 2018 | Body & Mind | High | Sexual abuse, physical abuse, emotional abuse, physical neglect, emotional neglect | 41(29 in meta-analysis) | 4680 | Not stated | • Sexual abuse and neglect affect severity of hallucination<br>• Sexual abuse, physical neglect, and emotional neglect are associated with delusion severity<br>• Sexual abuse affected severity of positive symptoms<br>• Emotional neglect and physical neglect are associated with severity of negative symptoms |
| Castellvi, P. | 2017 | Body & Mind | High | Child maltreatment | 26 | 143,730 | Netherland, New Zealand, United States, Norway, Canada, United Kingdom, Denmark, Finland | • Physical abuse increases risk of suicidal behaviour |
| Fusar-Poli, P | 2017 | Body & Mind | High | Childhood abuse, childhood neglect | 44 | Sample size varies by outcome (range 89 to 24,133) | Australia, Canada, Finland, Italy, Poland, South Korea, Switzerland, UK, USA, Turkey | • Ultra-high risk state for psychosis is associated with physical neglect, and emotional abuse |
| Jones, L | 2012 | Body & Mind | High | physical violence, sexual violence, emotional abuse, neglect and any combination of those | 17 | 18,374 | United States, United kingdom, Sweden, Finland, Spain, Israel | • Children with disabilities are at higher risk of physical, emotional and sexual abuse, and neglect. |
| Norman, R.E | 2012 | Body & Mind | High | Physical abuse, Emotional abuse, neglect | 124 | Sample sizes for individual meta-analyses not reported. Number of studies in meta-analyses range from 2–59 | Australia, New Zealand, Western Europe, North America | • Physical abuse, and emotional abuse increases risk for depressive disorder, anxiety disorders and eating disorders<br>• Physical abuse, and neglect doubled odds of childhood behaviour and conduct disorder<br>• Physical abuse, and neglect increased risk of alcohol misuse and dependence<br>• Physical abuse, emotional abuse, and neglect increased risk of suicidal behaviour<br>• Physical abuse, emotional abuse, and neglect were associated with increased risk of STI (including HIV) and increased risky sexual behaviour<br>• Physical abuse, emotional abuse, and neglect increased the risk of smoking and being obese |
| Winokur M. | 2014 | Social Factors | High | Abuse, Neglect | 102 | 666,615 | United States, Spain, Norway, Ireland, Israel, Sweden, the Netherlands, Australia | • Kinship care mediates the relationships between childhood abuse and mental health |

prevalence of violence exposure in disabled versus non-disabled children. There were high levels of heterogeneity due to type of reporting, study setting and type of disability.

Norman et al. [28] investigated a range of associations with health outcomes and physical abuse, emotional abuse and neglect. These consequences were not limited to mental health, and included HIV risk and obesity. However, most included papers were about mental health and substance use. The studies included in this review originated in Australia, New Zealand, Western Europe and North America. They found that adults who were physically abused (OR = 1.54, 95% CI 1.16–2.04), emotionally abused (OR = 3.06, 95%2.43–3.85) or neglected (OR = 2.11, 95% CI 1.61–2.77) were at higher risk of developing depressive disorders, anxiety disorders and eating disorders. The association between depression and physical abuse was only significant in high-income countries and not in low- and middle-income countries. However, the association between neglect and depression was the same across countries. Physical abuse and neglect were also associated with double the odds of developing behavioural and conduct disorders during childhood. Suicidal behaviour increased with exposure to physical and emotional abuse, as well as neglect. They also found a higher risk of alcohol misuse and dependence and to a lesser extent drug use. They found an increase in risky sexual behaviours and sexually transmitted infections (physical abuse OR = 1.78, 95% CI 1.50–2.10; emotional abuse OR = 1.75, 95% CI- 1.49–2.04; neglect OR = 1.57, 95% CI 1.39–1.78).There was an increased risk of current smoking associated with a history of emotional (OR = 1.70, 95% CI 1.55–1.87) and physical (OR = 1.55, 95% CI = 1.09–2.21) abuse; and an increased risk of obesity associated with physical (OR = 1.32, 95% CI 1.06–1.64) and emotional (1.24, 95% CI 1.13–1.36) abuse. The evidence for other associations with physical health problems, such as cardiovascular disease and cancer, was weak.

Winokur et al. [29] was the only high-quality paper not in the 'Body & Mind' category. The authors reviewed papers that compared outcomes for children removed from home due to abuse or neglect who were subsequently placed in kinship care (i.e. with extended family) versus non-kin foster care. 102 papers were included, with a total number of 666,615 children. Most of the included studies were conducted in the USA, with the rest conducted in Spain, Norway, Ireland, Israel, Sweden, the Netherlands and Australia. They reported that children placed in kinship care after suffering abuse or neglect had fewer behavioural problems (standardised mean difference = -0.33, 95% CI -0.49 to -0.17), fewer mental health disorders (OR = 0.51, 95% CI 0.42–0.62) and better wellbeing (OR = 0.50, 95% CI 0.38–0.64), than children placed in non-kin foster care.

The findings of these high-quality papers are mapped onto the bio-ecological model in Fig 5.

### Findings of medium-quality studies

There were 46 medium-quality papers. Within the 'Mind and Body' category 20 had the subtheme 'Mental Health and Substance Use and Misuse', eight had the subtheme 'Physical Health', and one had the subtheme 'Brain Structure, Neurodevelopment, Cognition and Personality'. In the category 'Social Factors', six papers explored the subtheme 'Relationships, Parenting, Sexual Behaviour' and four studied the theme 'Offending and Antisocial Behaviour'. 'Environmental Risk Factors' were investigated by two studies. The subthemes 'Resilience Factors' and 'Education/Adult Economic Status' had one study each. Thirty-two papers investigated more than one type of abuse or neglect, 12 concentrated on sexual abuse, one on physical abuse and one on emotional abuse. The quality rating of these studies was mainly influenced by the lack of a rigorous risk of bias assessment, or a failure to include the assessment outcome in the analysis of results.

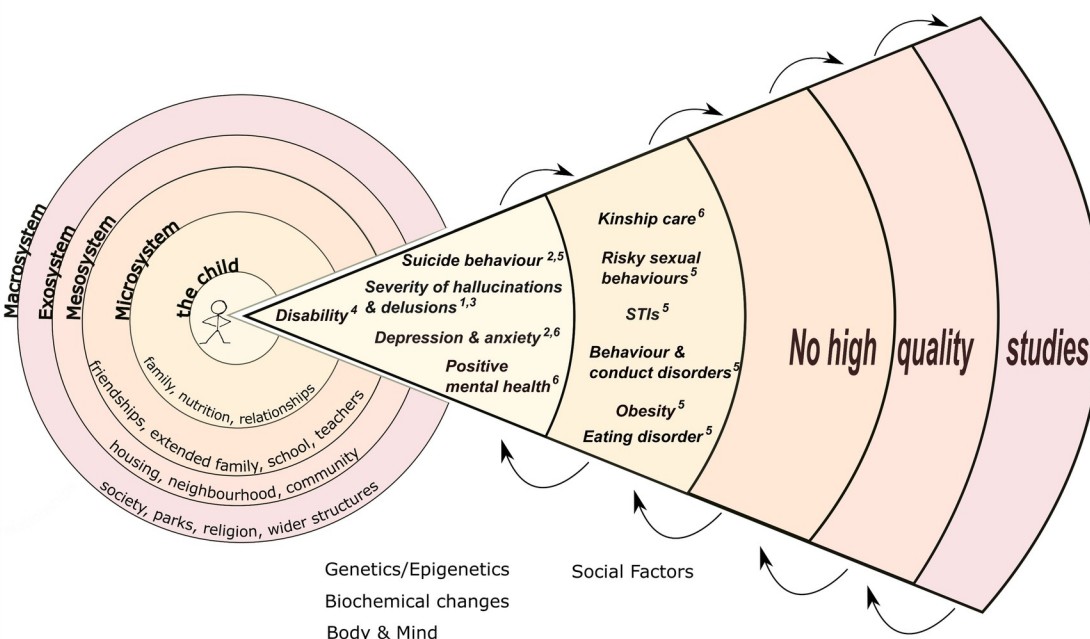

**Modelling of identified risk factors and the bio-ecological model.**

**Fig 5. Model of interactions of factors.** Key- 1) Bailey et al 2) Castellvi et al 3) Fusar-Poli 4) Jones et al 5) Norman et al 6) Winokur et al.

A summary of the papers, including suggestive findings is available in S1 Table.

## Discussion

From our review of the literature, we can confidently identify five hallmarks of CM.

1. Increased risk of psychopathology;

2. Increased risk of obesity;

3. Increased risk of participating in high risk sexual behaviours;

4. Increased risk of smoking;

5. Increased risk of CM in children with disabilities;

Yet despite the high quality of the meta-analyses that have identified these hallmarks, we still know nothing about the direction(s) of causality nor about the mechanisms underpinning them. For example, does psychopathology (or childhood temperamental factors leading to later psychopathology) increase the child's risk of experiencing CM or does CM increase the child's risk of psychopathology? Or both?.

There is already ample evidence for poorer physical and mental health outcomes for adults who have experienced CM; and smoking, obesity and even risky sexual behaviours could be mediators along these pathways. Certainly smoking and obesity are associated with a number of physical impairments and premature death [30], but mechanisms, confounders and causal pathways are unclear. For example, Attention Deficit Hyperactivity Disorder (ADHD) is associated with an increased risk of smoking [31], obesity [32] and CM [33], so would need to be

taken into account in any prospective longitudinal studies aiming to untangle these relationships.

There may be an argument that resilience could be regarded as a sixth hallmark. In all studies included in this review, there were participants who had been exposed to CM who had not developed the negative outcomes that were the focus of the study. However given that there were no direct high-quality studies looking at resilience per se it is not possible to say if these children had in fact experienced no long term negative outcome from their exposure to abuse and neglect, or if certain adverse outcomes which may have been present had simply not been measured. There is a need for researchers to consider designing high-quality studies which examine resilience directly as a carefully defined and measured outcome variable.

There is good evidence that having a disability is a risk factor for experiencing abuse and neglect [27]. This is an important focus for future research: it is often assumed that developmental problems are the *result* of abuse and neglect, but we have found this not to be the case, at least for symptoms of neurodevelopmental conditions such as ADHD and Autism [34,35].

The great majority of studies were conducted in WEIRD countries, making it challenging to understand community and societal underpinnings of the hallmarks. For example, there may be a mediating effect on the outcomes of CM in children who are placed in kinship (i.e. extended family) care versus statutory foster care [29]. However, the role of extended families versus non-relative foster families in the care of maltreated children varies so greatly across the world [36], that this finding sheds little light on mechanisms. Further research in more diverse populations will be crucial.

Our findings are notable for the absence of high-quality systematic reviews or meta-analyses focusing on biological mediators such as epigenetics, stress reactivity, immune function, and brain structure and function. Due to the methodology employed in this study, this absence may arise from either deficits in the primary literature, or an absence of high-quality reviews. Regardless, this highlights the importance of consolidating and improving our understanding of these important areas of study.

Despite the burgeoning number of studies on the effects of CM, there remains a fundamental issue with the quality of much of the literature, across both systematic reviews and meta-analysis. Of the 178 studies included in this review only 3% were rated as high quality using the AMSTAR-2 tool and only a further 27% managed a moderate quality rating.

There is currently no agreed standard with relation to how studies report their exposures and outcomes. For example, in considering the types of CM (or adverse childhood experiences more widely) that study participants have been exposed to, some authors report this precisely, allowing for replication in further studies, however many do not. This makes synthesis of outcome findings challenging if not impossible and decreases the likelihood of findings emerging consistent with sub-types of CM. Adoption of an agreed standard in terms of the reporting of exposure to CM in study participants, and of commonly measured outcomes, would help increase the quality of future meta-analysis, and perhaps make possible a network study which could help unravel the complexity of the underlying interactions between variables.

A major oversight in the extant research on CM is the fact that we were able to find no review linking different factors across domains or considering multiple levels of the bio-ecological model. The potential for interactions between factors across domains is therefore not addressed at all despite the large number of "silo" studies reviewed here. A high level of complexity is inevitable when biological systems relevant to CM have such diverse purposes, components and actions, yet are intimately related in their functioning—as is true, for example, for the HPA-axis and the immune system. Methodologies adapted for complex systems are therefore crucial if we are to advance in this field. For more information on this see Ioannidis and colleagues [11]. Questions such as "how do physical factors affect mental health factors?" are

not considered at all in these reviews of the literature. Of the papers reviewed, only one high-quality review looked at mediating or moderating factors (kinship care) that might link CM with outcomes.

We found no high-quality reviews considering the potential impact which social relationships (either positive or negative) might have on the manifestations of effects of CM. This may be a challenging area in which to work as social relationships could be seen as cause, confounder and outcome. The same would be true of other outer aspects of the Bio-Ecological model such as the impact of social policy and state actions. We can make no comment on the effects of the outer layers of the bio-ecological model since there is virtually no evidence available, here, at present. This area has not been studied in detail and requires further consideration from researchers.

Examining the gaps, there is clearly a need for future researchers in this field to consider study designs that embrace complexity if crucial unanswered questions, especially about causality and mechanisms, are to be addressed. This is no truer than around the question of resilience. Given the lack of focus on resilience in the reviews we have examined, we are not able to answer any questions in relation to how to prevent adverse outcomes in children exposed to CM. This is an area of research which we would argue requires urgent attention.

Finally, there were no high-quality reviews which reliably addressed the potential significance of the timeline of exposure to CM in relation to the developmental stage of the child. For example, questions have not been answered regarding whether there are ages or stages of development which are particularly sensitive to the risk for the development of certain outcomes of CM.

This study aimed to elucidate hallmarks of CM robustly evidenced across multiple high-quality studies. In terms of limitations, our hallmarks are confined to human studies rather than across taxa as in the hallmarking work on aging and cancer. Whilst there are animal models of early life stress, we were looking more specifically at effects of abuse and neglect which is not readily distinguished from other sources of early stress in animal models. Secondly, our conclusions are based on the quality of systematic review articles and meta-analyses rather than on the underlying primary research. There might be undetected hallmarks based on high-quality individual studies that we missed because they have not been subjects of systematic reviews or because the systematic review was of low or moderate quality. Our search was limited to articles in English, and by limiting our search to systematic reviews we may have omitted relevant findings in the "grey literature". We did consider undertaking a network analysis however this was not possible due to the heterogeneity of outcomes and study parameters. Indeed, this heterogeneity may also have impacted our identification of hallmarks since it is likely to have limited the potential for meta-analysis.

## Conclusions

We believe that we have, for the first time, demonstrated five hallmarks of exposure to CM: *Increased risk of psychopathology; Increased risk of obesity; Increased risk of high- risk sexual behaviours, Increased risk of smoking;* and *that the risk of CM exposure is increased in children with disabilities.* It may be that resilience represents a sixth hallmark however further research is required to confirm this.

In our "review of reviews" we identified significant absences of high-quality reviews in important areas such as biological factors and wider societal factors such as the quality of neighbourhoods. These gaps must be addressed if progress is to be made in understanding the impact, and mechanisms of impact, of CM and, more importantly, understanding how to protect abused and neglected children from adverse outcomes.

Using study designs that embrace complexity, in order to examine inter-relationships within and across the bio-ecological model, is likely to be key in answering some of these outstanding questions. Future studies need to be adequately designed and powered to achieve this.

## Supporting information

**S1 Table. Details of medium-quality articles.**
(PDF)

**S1 File. PRISMA checklist.**
(PDF)

**S2 File. Detailed search strategy.**
(PDF)

**S3 File. AMSTAR-2 items.**
(PDF)

## Acknowledgments

We are grateful for the contributions of Evi Bali, Encrico Venturini, Emily Roberts, Pablo Barrera, Lilliane Bills, Makhib Choudkhuri, Orla Macpherson, and Rachel Whyte to the double rating of papers. We are also grateful to Irene O'Neill for administrative support.

## Author Contributions

**Conceptualization:** Jason Lang, Daniel M. Kerr, Paul Shiels, Helen Minnis.

**Data curation:** Jason Lang, Daniel M. Kerr, Papoula Petri-Romão.

**Formal analysis:** Tracey McKee.

**Methodology:** Jason Lang, Daniel M. Kerr, Papoula Petri-Romão, Tracey McKee, Helen Smith, Naomi Wilson, Marianna Zavrou.

**Project administration:** Jason Lang, Papoula Petri-Romão, Tracey McKee.

**Supervision:** Paul Shiels, Helen Minnis.

**Visualization:** Papoula Petri-Romão.

**Writing – original draft:** Jason Lang, Daniel M. Kerr, Papoula Petri-Romão.

**Writing – review & editing:** Jason Lang, Daniel M. Kerr, Tracey McKee, Helen Smith, Naomi Wilson, Marianna Zavrou, Paul Shiels, Helen Minnis.

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
