## [Decision Letter · Decision Letter 0]

15 Jun 2020

PONE-D-20-08783

The hallmarks of childhood abuse and neglect: a systematic review

PLOS ONE

Dear Dr. Kerr,

Thank you for submitting your manuscript to PLOS ONE. After careful consideration, we feel that it has merit but does not fully meet PLOS ONE’s publication criteria as it currently stands. Therefore, we invite you to submit a revised version of the manuscript that addresses the points raised during the review process.

We look forward to receiving your revised manuscript.

Kind regards,

Abraham Salinas-Miranda

Academic Editor

PLOS ONE

Reviewers' comments:

Reviewer's Responses to Questions

**Comments to the Author**

1. Is the manuscript technically sound, and do the data support the conclusions?

Reviewer #1: Partly

Reviewer #2: Yes

2. Has the statistical analysis been performed appropriately and rigorously? 

Reviewer #1: N/A

Reviewer #2: N/A

3. Have the authors made all data underlying the findings in their manuscript fully available?

Reviewer #1: Yes

Reviewer #2: Yes

4. Is the manuscript presented in an intelligible fashion and written in standard English?

Reviewer #1: Yes

Reviewer #2: Yes

5. Review Comments to the Author

Reviewer #1: This manuscript undertook the laudable goal of combing the entirety of the literature on child maltreatment to identify the “hallmarks” with which it is associated. This is clearly a large and rigorous undertaking. However, I am left with the question, what does this add? All of the reported hallmarks are already well established in the CM literature so it is unclear to me what the unique contribution is here. I think it would strengthen the manuscript to really consider and make clear to the reader the new knowledge gained. I think the authors posed a number of important points about unanswered questions pertaining CM in the introduction but none of this issues materialized. To me, the gaps in the literature that were identified from this review is perhaps more important than the hallmarks that were identified. I would argue that the authors could focus less on what they did find, and use the results to address what was not found.

I have some additional comments below.

Make clear in Abstract and through the paper whether you are referring to hallmarks associated with child maltreatment (CM) perpetration or victimization. Also, whether the hallmarks represent risk factors for, or consequences of CM.

The authors begin the introduction with a discussion of ACEs and seem to use this concept interchangeably with CM. However, ACEs are far broader than just CM which falls under the umbrella of ACEs. The authors should avoid conflating them and restrict the literature review to CM.

The introduction devotes a lot of space to the importance of looking at biological hallmarks for CM but none of this is reflected in the results. I realize that this is due to it being a gap in the literature. However, it seems misleading to frame this being across the bio-ecological spectrum when there is no data on this issue.

Related to this point, the authors lay out a series of unanswered questions about CM in the introduction (see below); however, none of these questions are addressed in hallmark review.

“in abused and neglected individuals, what are the associations between biological factors (e.g. epigenetic modifications of DNA) and psychological factors (e.g. suicidality)? What is the impact of the social world of the child on risk and resilience in the context of abuse and neglect? At which developmental periods are abuse and neglect most likely to increase risk of negative mental and physical health outcomes? What is the impact of different forms of abuse at different critical periods in development? Does this differ by gender?”

I think it would be helpful to the reader to provide more detail l about how AMSTAR assesses quality

Reviewer #2: The articles of review: “The hallmarks of childhood abuse and neglect: a systematic review,” provides both solid methodological reasoning and application as well as interesting findings. The area of child maltreatment has been studied across various fields, leading to the importance of this article. The authors provide clarity to the current spread of reviews across fields and consolidate their findings using a useful and little-known methodology in the field of social sciences (Hallmarking). Overall, the article provides both a solid report of their methods as well as the outcome of the study. Although, there are a few areas that should be address related to some areas of confusion related to sentences that need simplifying and restructured. Additionally, there are some areas of concern related to the need for additional details on some of the findings as well as potential mislabeling of figures. The details of these concerns are outlined in the bullet points below, additional details are provided in the attached PDF with highlights and comments.

•Need for general proofreading of the paper, as spelling/wrong words for the sentence were found throughout the draft.

• Consider removing or consolidating the rhetorical questions on page 3

• Consider rewriting, simplifying or adding clarity to some of the sentences. They are indicated in the PDF draft

• Need to adjust some of the writing structure for clarity such as page 10- indicated in the PDF

• Need details on subthemes of the table provided in the article surrounding high quality articles. I would consider referencing to a blank version of figure 5 during your breakdown of subthemes as the image helps clarify the sub themes. You would have to renumber/name your figures.

• Need to have details on the rating scale for quality of studies. This is especially important when there is meta-analysis being stated as ranked high but lack information on sample size, countries of studies etc. It should be explained how according to the protocol these are considered highly ranked.

• Issues with font colors in figure 5. I would keep in mind the resolution that images will be in the final publication. The grey and white text on a light background will be difficult for some readers to read clearly.

• I think you potentially have your figures misnamed in the text. Is the figure your referring to in “Figure 4. Model of interactions of factors. Key- 1) Bailey et al 2) Castellvi et al 3) Fusar-Poli 4) Jones et al 5) Norman et al 6) Winokur et al. Is this really referring to fReally figure 4 which is a map or figure 5?

• In your conclusion, you make the statement “ There are clear gaps in the literature, for example there is little research on certain biological factors and virtually no research on wider societal factors such as the quality of neighborhoods.” I would be careful with this statement as from what I understand of your study. Your team conducted a review of reviews and therefore as you stated in your limitations the team could had missed literature.

• In the supplementary information S1- I am confused why there is no protocol indicated ? Could the authors explain why this is not indicated?

6. PLOS authors have the option to publish the peer review history of their article (what does this mean?). If published, this will include your full peer review and any attached files.

Reviewer #1: No

Reviewer #2: No

---

## [Author Response · Author response to Decision Letter 0]

12 Aug 2020

Reviewer 1: 

We agree that our negative findings are likely to be of greater significance than the positive findings and have made changes to the content and emphasis of our discussion to address this. We also agree that questions raised in the introduction are not fully addressed by the review. Parts of the introduction have been re-written in response to other comments and as such some of these questions are no longer present; and where they are, we have been more explicit in addressing the inability of our review to answer them in our discussion. 

In reference to further comments made: 

1) The manuscript has been changed to use more consistent terminology around child maltreatment and to be more explicit around whether hallmarks are consequences or risk factors for CM. For clarity we are only addressing hallmarks of exposure to CM (be they risk factors for being exposed to CM, or consequences of exposure to CM). Perpetration of CM is only of relevance to our review if it is being studied as a consequence of CM (eg. are individuals exposed to CM at greater risk of perpetrating CM themselves?)

2) The introduction has been changed to be more explicit in our focus on CM and to explicitly distinguish CM from the wider concept of ACEs. 

3) The introduction has been re-written in response to point 2 and as such some of these original points are no longer included. Where these questions are included we have been more explicit in the discussion about how our review was not able to answer these. 

4) We have added a more detailed description of AMSTAR-2 in the methods section and have included the items included in AMSTAR-2 as a supplementary material. 

Reviewer 2: 

1) We are grateful for your comments on the draft and for highlighting some unfortunate typographical errors. Parts of the manuscript have been re-written in response to other comments; where the highlighted errors persist we have corrected these. 

2) We have included more details on the sub-themes in our results sections and referenced to a blank version of the bio-ecological model as suggested. 

3) We have included a more detailed description of AMSTAR-2 in the methods section and added the items included in AMSTAR-2 as a supplementary material. In reference to the high-quality studies, AMSTAR-2 assigns a score of high-quality to studies with no, or one non-critical weakness. Not describing the countries of origins of included papers would be considered a non-critical weakness and such papers could be rated as high quality provided that no other weaknesses are present. The paper by Fusar-Poli included a range of sample sizes for individual analyses but did not give an overall sample size. The table has been amended to make this clearer. 

4) The figure has been adapted to aid clarity. 

5) The figures have been re-numbered. Thank you for pointing out this error.

6) We agree that all our findings must be interpreted in light of the specific methodology that has been employed. We have amended the specific phrase highlighted, and overall emphasis of the discussion in an effort to make this clearer. 

7) An internal protocol was developed but was not published or registered. The supplementary materials have been amended to make this clear.

---

## [Decision Letter · Decision Letter 1]

25 Nov 2020

The hallmarks of childhood abuse and neglect: a systematic review

PONE-D-20-08783R1

Dear Dr. Kerr,

We’re pleased to inform you that your manuscript has been judged scientifically suitable for publication and will be formally accepted for publication once it meets all outstanding technical requirements.

Kind regards,

Abraham Salinas-Miranda, MD, PhD

Academic Editor

PLOS ONE

Additional Editor Comments (optional):

The paper has improved significantly and contributes to the literature with robust findings about factors that are associated to child maltreatment. The authors clarified their use of PRISMA and AMSTAR and fixed their errors satisfactorily.

Reviewers' comments:

Reviewer's Responses to Questions

**Comments to the Author**

1. If the authors have adequately addressed your comments raised in a previous round of review and you feel that this manuscript is now acceptable for publication, you may indicate that here to bypass the “Comments to the Author” section, enter your conflict of interest statement in the “Confidential to Editor” section, and submit your "Accept" recommendation.

Reviewer #2: All comments have been addressed

Reviewer #3: All comments have been addressed

2. Is the manuscript technically sound, and do the data support the conclusions?

Reviewer #2: Yes

Reviewer #3: Yes

3. Has the statistical analysis been performed appropriately and rigorously? 

Reviewer #2: Yes

Reviewer #3: Yes

4. Have the authors made all data underlying the findings in their manuscript fully available?

Reviewer #2: Yes

Reviewer #3: Yes

5. Is the manuscript presented in an intelligible fashion and written in standard English?

Reviewer #2: Yes

Reviewer #3: Yes

6. Review Comments to the Author

Reviewer #2: Thank you for your full integration of comments in this draft. It has greatly helped in providing clarity in sections of concern in my previous comments.

Reviewer #3: I was asked to review the article "The hallmarks of childhood abuse and neglect: a systematic review".

The topic is original and addresses an important public health issue, which is child abuse.

The authors applied PRISMA and AMSTAR-2 to guide the methods of their study and the bioecological model. I do not find any issues with their application of AMSTAR in the current version of the manuscript.

They found hallmarks of child abuse, which represent common factors across studies with most robust findings. These were: 1. Increased risk of psychopathology;

2. Increased risk of obesity;

3. Increased risk of participating in high risk sexual behaviours;

4. Increased risk of smoking;

5. Increased risk of CM in children with disabilities;

Their conclusions are adequate based on their findings and goal of hallmarking. They have noted their limitations as a review paper.

Figure and tables have been fixed.

The paper was improved significantly from the previous versions and is ready to move forward in my opinion.

7. PLOS authors have the option to publish the peer review history of their article (what does this mean?). If published, this will include your full peer review and any attached files.

Reviewer #2: No

Reviewer #3: Yes: Abraham Salinas-Miranda, MD, PhD

---

## [Editor Report · Acceptance letter]

27 Nov 2020

PONE-D-20-08783R1 

The hallmarks of childhood abuse and neglect: a systematic review 

Dear Dr. Kerr:

I'm pleased to inform you that your manuscript has been deemed suitable for publication in PLOS ONE. Congratulations! Your manuscript is now with our production department. 

Kind regards, 

on behalf of

Dr. Abraham Salinas-Miranda 

Academic Editor

PLOS ONE